# Abrupt Gulf Stream path changes are a precursor to a collapse of the Atlantic Meridional Overturning Circulation
René M. van Westen ✉ & Henk A. Dijkstra

The Gulf Stream is part of the Atlantic Meridional Overturning Circulation (AMOC). The AMOC is a tipping element and may collapse under changing forcing. However, the role of the Gulf Stream in such a tipping event is unknown. Here, we investigate the link between the AMOC and Gulf Stream using a high-resolution (0. 1°) stand-alone ocean simulation, in which the AMOC collapses under a slowly-increasing freshwater forcing. AMOC weakening gradually shifts the Gulf Stream near Cape Hatteras northward, followed by an abrupt northward displacement of 219 km within 2 years. This rapid shift occurs a few decades before the simulated AMOC collapse. Satellite altimetry shows a significant (1993–2024, $p < 0.05$) northward Gulf Stream trend near Cape Hatteras, which is also confirmed in subsurface temperature observations (1965–2024, $p < 0.01$). These findings provide indirect evidence for present-day AMOC weakening and demonstrate that abrupt Gulf Stream shifts can serve as early warning indicator for AMOC tipping.

The Gulf Stream (GS) is a well-studied western boundary current system and is part of the upper and northward flowing branch of the Atlantic Meridional Overturning Circulation (AMOC)[1-4]. The GS detaches from the continental margin near Cape Hatteras (35.1°N, 75.3°W) and flows into the open ocean, also known as the Gulf Stream Extension[5-7]. Variations in the GS path can be linked to global climate variability[8], changing atmospheric forcing[9-11], and changing AMOC strength[12-14]. This AMOC-GS connection is noteworthy, as the GS path is influenced by both the northward and southward flowing branches of the AMOC[15,16]. In particular, the GS path is strongly shaped by deep flow interactions with the bathymetry[17,18]. Most AMOC- and GS-related observations are conducted in the upper ocean[19], and changes in the GS path could thus provide insights into deeper ocean circulation changes.

The AMOC is considered a tipping element in the climate system[20,21], implying that the present-day AMOC, with a strength of about 17 Sverdrups (1 Sv = $10^6$ m$^3$s$^{-1}$,[22,23]), can substantially reduce in strength (<5 Sv) under future climate change[24,25]. Abrupt changes in AMOC strength can drive shifts in the GS path, with paleo-climatic evidence reported from the study of Dansgaard-Oeschger events[26]. Depending on the lead-lag relationship between GS path shifts and AMOC changes, GS shifts could, in principle, serve as an early warning indicator of a forthcoming AMOC collapse. This information would be valuable to society, as a substantially weakened AMOC can cause large-scale climate shifts[27-30], with Europe particularly at risk of drastic changes[31-33].

So far, the AMOC is likely weakening, and reconstructions based on sea surface temperatures suggest about a 15% reduction in its strength since 1950[14,34,35]. GS path changes have been reported over the past decades, but the reported shifts are not consistent[36-38]. For example, the GS path derived from subsurface temperatures shifted by 0.45°N between 75°W and 50°W and over the period of 1965–2017[9]. However, the GS path from satellite altimetry since 1993 shows northward changes between 75°W and 70°W, while shifting southward further downstream[39]. Intermodel differences are also found, where the GS path shifts either northward or southward under increasing AMOC strength[12,40]. These differences are attributed to different GS path definitions (surface versus subsurface[9]) and a limited GS representation due to a relatively coarse horizontal resolution[12]. Alternatively, it is possible that the current AMOC weakening is still too small to induce robust GS path changes and that the observed GS changes are primarily related to atmospheric responses[41,42].

The relation between GS path shifts and AMOC changes can be tested using climate model simulations in which the AMOC substantially weakens under an external forcing, such that there is a large signal-to-noise ratio. Assessing this AMOC-GS coupling poses several challenges, particularly when the applied freshwater flux forcing is too strong[27,28,43] or when it is difficult to disentangle AMOC-induced shifts from those caused by transient climate change[24,25,44]. The most significant limitation, however, is the horizontal ocean resolution (typically 1° in most models used in the Climate Model Intercomparison Project, phase 6), which is too coarse to accurately

Department of Physics, Institute for Marine and Atmospheric research Utrecht, Utrecht University, Utrecht, the Netherlands. ✉e-mail: r.m.vanwesten@uu.nl

represent mesoscale processes and western boundary current dynamics[7,45–48]. For example, the quasi-equilibrium hosing simulation conducted with the Community Earth System Model (CESM) using a 1° ocean component[49] is not suitable for this purpose, as the GS path is not well captured.

Recently, a quasi-equilibrium hosing simulation was performed using the high horizontal resolution (0.1°) stand-alone ocean component of CESM, the Parallel Ocean Program (POP, version 2)[50]. An additional advantage of using POP is that GS path variations are solely driven by ocean circulation (i.e., AMOC) changes. The atmospheric forcing is prescribed with seasonally repeating near-surface atmospheric temperatures and bulk formula (i.e., surface wind stress); hence, atmospheric-induced GS path changes[41,42,51] cannot be captured. In the POP, the AMOC collapses under a slowly and linearly increasing freshwater flux forcing, which was applied over the North Atlantic Ocean (see "Methods"). Here, we report on the coupled AMOC-GS behavior in this POP simulation.

## Results

### Shifts in the Gulf Stream Path

We first focus on 26.5°N because the GS (here, the Florida Current) and AMOC have been monitored along the RAPID-MOCHA array since 2004[52]. The Florida Current strength at 26.5°N is 27.3 Sv during the first 50 model years in POP (Fig. 1a, b), which is somewhat lower (by 14%) than the observed Florida Current strength of 31.7 Sv[19]. The AMOC at 26.5°N over the upper 1000 m has an initial strength of 19.9 Sv (first 50 model years), which is 17% stronger than the observed value of 17 Sv[22,23]. Under the increasing freshwater flux forcing, the AMOC begins to collapse around model year 420 and equilibrates over the last 50 model years[50]. Over these last 50 model years, a residual and relatively weak AMOC strength of 5.1 Sv remains. This is the collapsed AMOC state in POP, as there is no deep water formation at the higher latitudes nor northward meridional heat transport

over all Atlantic Ocean latitudes[50]. Hence, we primarily compare the first and last 50 model years in our analysis. The Florida Current strength weakens as well over the first 500 model years, with a time-mean strength of 20 Sv over the last 50 model years.

The AMOC collapse is also evident when analysing the meridional velocity responses along 26.5°N (Fig. 1c) and the weakening in the Deep Western Boundary Current (DWBC; Fig. 1d). DWBC observations indicate a strength of about 32 Sv[53], and the DWBC in POP has an initial strength of 37.9 Sv (first 50 model years), which reduces to 16.5 Sv (last 50 model years). The AMOC strength below 1000 m depth starts at 21 Sv and declines to 5 Sv. The POP results (first 50 model years) can also be compared against the high-resolution reanalysis product GLORYS12V1 (1/12°, 1993–2024), with volume transports close to observations, showing a reasonably good agreement (Fig. S1).

The substantially weaker AMOC induces dynamic sea-level (DSL) rise over the North Atlantic Ocean in the POP[50,54], which is also shown along 26.5°N in Fig. 1a, c (upper panels). On top of the basin-scale DSL changes, a weaker Florida Current reduces the zonal DSL gradient near the eastern coast of Florida[55], resulting in a total DSL rise of 66 cm near 80°W. Quantifying the GS path using the fixed 25 cm DSL isoline[11,56] is therefore not very useful in this POP simulation. Alternatively, the subsurface (200 m depth) 15 °C isotherm, referred to as the GS north wall[9,12], can be used to quantify the GS path. A drawback of this procedure is that oceanic temperatures are influenced by the reduced heat transport from a weaker AMOC. Hence, we quantify the GS path by first determining the upstream DSL value at the maximum zonal DSL gradient along 26.5°N (see stars in insets Fig. 2a, b). This DSL value is then followed downstream and is independent of the background DSL fields. The outlined procedure accurately works for monthly-averaged DSLs and (multi) yearly-averaged DSLs, which show a varying GS path between 75°W and 50°W (Fig. 2a, b). The GS separates from the continental margin near Cape Hatteras in the first 390 model years

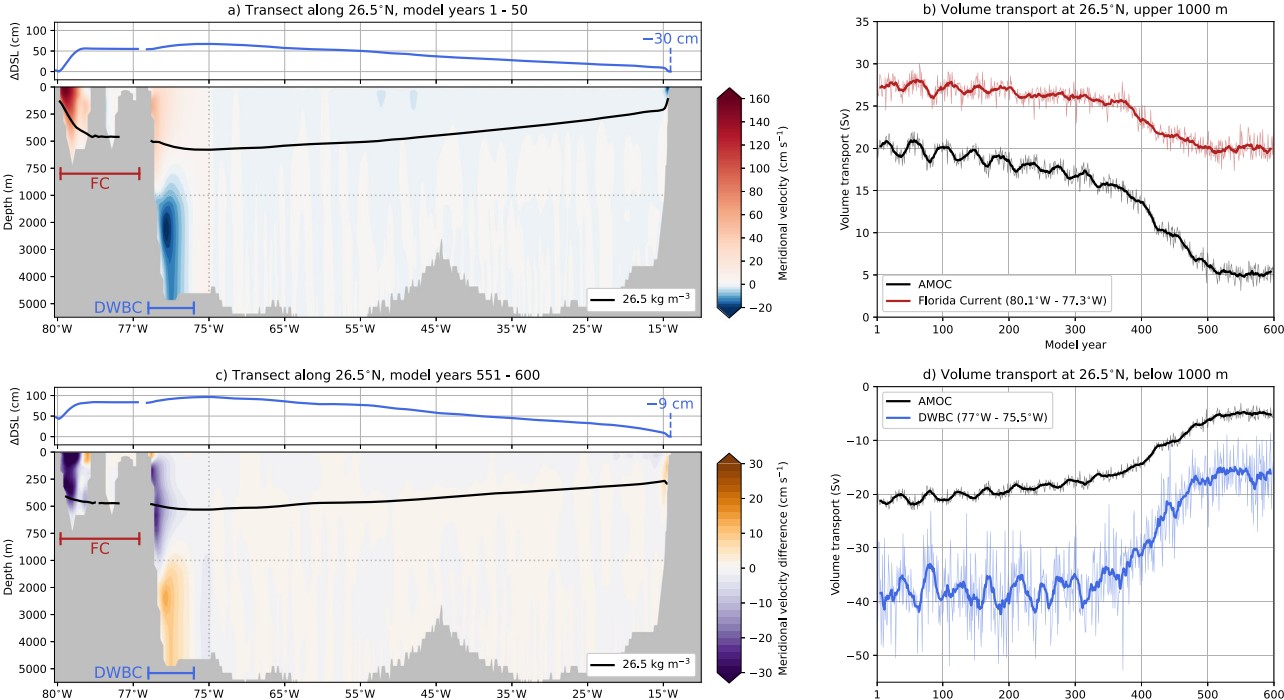

**Fig. 1 | Oceanic Responses along the 26.5°N transect. a, c** The dynamic sea-level (DSL) difference compared to the east coast (including its value), the meridional velocity, and the $\sigma_0 = 26.5$ kg m$^{-3}$ depth along the 26.5°N transect for model years **a** 1–50 and **c** 551–600. In **c**, only the meridional velocities for model years 551–600 are displayed as the differences to model years 1–50. Note the different horizontal and vertical spacing for the axes (**b, d**): The volume transports over the upper 1000 m

and below 1000 m for the AMOC strength (west to east coasts), Florida Current (FC, 80.1°W–77.3°W, upper 1000 m) and the Deep Western Boundary Current (DWBC, 77°W–75.5°W, below 1000 m), the locations of which are indicated in (**a, c**). The thin curves are yearly-averaged volume transports, whereas the thick curves are smoothed versions (11-year moving window).

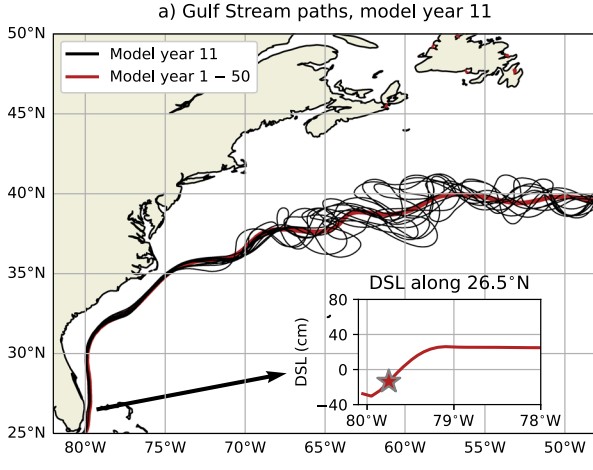

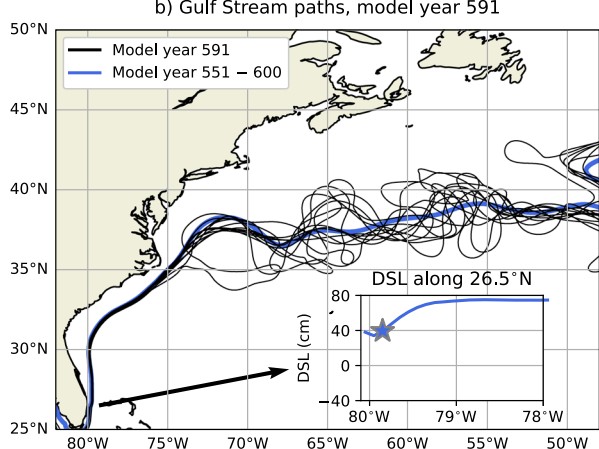

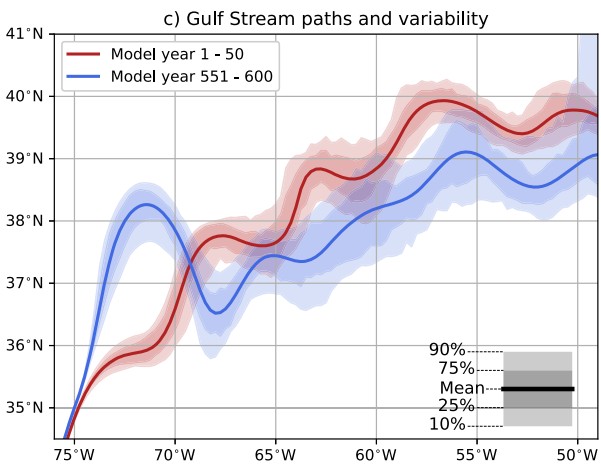

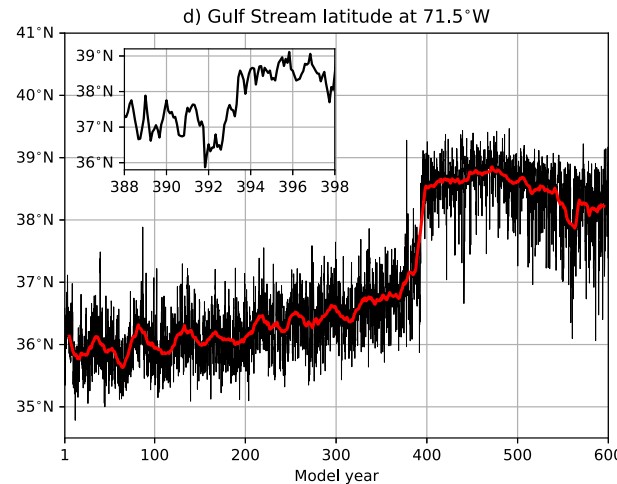

**Fig. 2 | Gulf Stream path changes during AMOC collapse. a** The monthly-averaged GS paths for model year 11, including the time-mean GS path for model years 1–50. For the latter, first, the time-mean DSL is obtained, and then the GS path is determined. The GS paths are derived from DSL isolines starting at 26.5°N, where the DSL at the maximum zonal DSL gradient is used (the stars in the inset) for the GS path. **b** Similar to panel a, but now the monthly-averaged GS paths for model year 591 and time-mean GS path for model years 551–600. **c** The GS paths for model years 1–50 and model years 551–600, where first the time-mean DSL field is obtained and then the GS path is determined. The shading represents the GS path variability using 50 different years, where first the yearly-averaged DSL field is obtained and then the GS path is determined. **d** The monthly-averaged GS latitude at 71.5°W, including a 11-year moving average (red curve). The inset shows a zoomed-in version from model year 388–398.

of the POP, which contrasts with most climate models, where the GS detaches farther north[57,58].

There is a strong response in the GS path between the first and last 50 model years, with the GS shifting northward between 75°W and 69°W, and southward between 69°W and 50°W (Fig. 2c). The largest meridional shift of 2.35° (261 km) occurs at 71.5°W. At this longitude, the GS latitude gradually shifts northward by about 1.2° (133 km) over the first 392 model years, followed by an abrupt northward shift of 1.97° (219 km) between model years 392 and 394 (Fig. 2d and Figure S2). For this northward shift we used the GS path from yearly-averaged DSL fields, as the seasonal cycle may influence the monthly-averaged GS path (inset in Fig. 2d). When varying the starting year (382–392) and ending year (394–404) to quantify the GS path shift, we find a mean northward shift of 1.38° (153 km) over all combinations, with 0.89° to 2.03° for the 95% CI (2.5th to 97.5th percentiles). The standard deviation is 0.245° for the GS latitude at 71.5°W for model years 360 to 390 (yearly-averaged GS paths, any trend was removed), which implies that the abrupt shift in the mean is a factor of 5.6 larger than the standard deviation, and a factor of 3.6–8.3 for the 95% CI. The GS north wall shows very similar meridional shifts compared to the GS path (compare Figs. S2j and 2c). There is also an abrupt transition for the GS north wall at 71.5°W (insets in Figs. S2 and S2k), and it is not related to how the GS path is determined.

The GS destabilisation point, i.e., the point where the GS transitions to an unstable and strongly meandering jet (see Methods), remains fairly constant over the first 392 model years with a time mean of 66.3°W (Fig. S2l), which agrees well with observations[59]. After model year 392, there is a mean westward shift of 2.84° (250 km) between model years 382–392 and 394–404, which is a factor of 1.6 larger than its standard deviation (of model years 360–390). The GS destabilisation point is not sensitive under AMOC weakening, and hence it is more useful to analyse the GS latitude at 71.5°W to find indications of AMOC weakening.

Most relevant here is that this GS path transition occurs prior to the onset of the AMOC tipping event. The different GS paths may be related to different GS path equilibria, which have been found in shallow-water models[60,61]. A gradually shifting GS path has been found in observations and was suggested to be linked to a weaker DWBC[39,62].

## Changes in the deep western boundary current
Although the relation between the DWBC and the AMOC is quite complicated, process modeling studies have indicated that a weaker DWBC can shift the GS separation point northward[14,15]. In these studies, a sufficiently strong DWBC induces bottom vortex stretching at the crossover point with the GS and, from vorticity conservation, the GS detaches from the

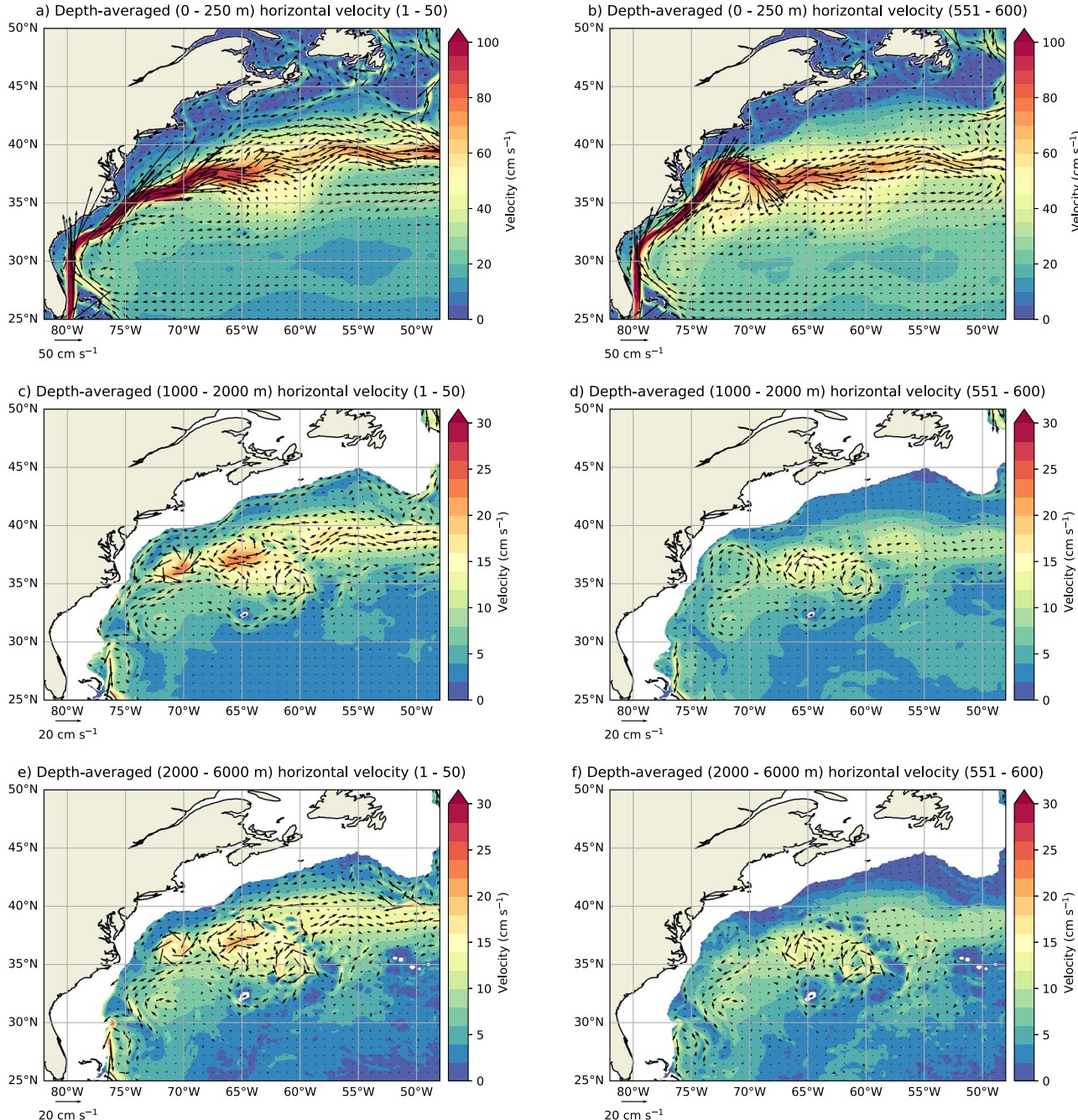

**Fig. 3 | Horizontal velocities responses.** The depth-averaged horizontal velocities over the upper 250 m (**a**, **b**), between 1000 m and 2000 m depths (**c**, **d**), and between 2000 m and 6000 m depths (**e**, **f**), for model years 1–50 and 551–600. The velocity is represented as $\sqrt{\overline{U}^2 + \overline{V}^2}$, where $U$ and $V$ are the depth-averaged zonal and meridional velocity, respectively, and the bars indicate time means. The quivers represent the time-mean zonal ($\overline{U}$) and meridional ($\overline{V}$) velocity. Note the difference in the ranges of the colourbars and quivers.

continental margin[14,17]. The DWBC strength at 26.5°N in POP is indeed decreasing (Fig. 1d) as the GS shifts northward between 75°W and 69°W (Fig. 2c). However, to better understand the GS-AMOC interaction, it is more useful to analyse the upstream DWBC (from the crossover point), which is shown here as the horizontal velocities over three different depth layers over the Gulf Stream Extension region (Fig. 3).

In the upper 250 m, strong ocean currents of more than 1 m s⁻¹ are found, which are associated with the GS (Fig. 3a, b). The meridionally shifting GS is clearly visible when comparing the first and last 50 model years. Another notable difference is the almost complete cessation of the (downstream) Labrador Current south of Newfoundland and Nova Scotia.

The (upstream) Labrador Current (in the Labrador Sea) does not collapse (not shown). In the upper DWBC core, which we represent here as the horizontal velocities between 1000 m and 2000 m depths (Fig. 3c, d), the DWBC follows the entire continental slope down to 26.5°N prior to the AMOC tipping event. Thereafter, the DWBC almost vanishes, meaning that it hardly induces any bottom vortex stretching at the crossover point with the GS, explaining the northward shifting GS. In the lower DWBC core, shown here by the horizontal velocities between 2000 m and 6000 m depth (i.e., the bottom, Fig. 3e, f), there is also a DWBC contribution along the continental slope. However, it is about a factor of 3 smaller compared to the 1000–2000 m layer and reduces to zero after the AMOC collapse.

To further quantify the DWBC changes, we analyse the zonal velocity responses along 60°W (Fig. 4a, b); any other longitude across the Gulf Stream Extension gives a similar result. The 60°W section is chosen because the upstream DWBC is hardly affected by the GS, and the Labrador Current has relatively large zonal velocities here. At this longitude, the GS shifts southward by 0.63° (70 km), which is also reflected in increased zonal velocities at 37.5°N and over the upper 250 m. The slope of the 26.5 kg m$^{-3}$ isopycnal decreases between 37.5°N and the continental shelf, indicating a weakening of the GS. Ocean currents flow westward along the continental slope (≈43°N), with the upper 1000 m primarily influenced by the Labrador Current and depths below 1000 m representing the DWBC. The oceanic state in POP (first 50 model years) is again in good agreement with that in the GLORYS12V1 reanalysis product (Fig. S3a–d).

Because the upper DWBC core (1000–2000 m) gives a relatively large response along the continental slope (Fig. 3c, d), we show the depth-averaged zonal velocities along 60°W over the same depth range in Fig. 4c. Near 43°N, the westward velocities gradually decline from about 6 cm s$^{-1}$ (first 50 model years) to near zero values after model year 500. This demonstrates there is hardly any bottom vortex stretching at this longitude, which is also the case downstream and closer to the GS crossover point (Figs. 3 and S4).

We also quantify the volume transports by the DWBC at 60°W, where we consider the westward ocean currents close to the continental slope between 39.5°N and 43.5°N (Fig. 4d). The DWBC strength is 60.3 Sv in the first 50 model years and reduces to 4.7 Sv in the last 50 model years. Note that this DWBC strength at 60°W is substantially stronger than the downstream DWBC at 26.5°N, meaning that part of this volume transport is recirculated over the Gulf Stream Extension. The initial DWBC in POP is stronger (+36%) compared to the GLORYS12V1 with a time-mean strength of 44.3 Sv (Fig. S3e). A part of the DWBC does not contribute to bottom vortex stretching, as there is hardly any slope in the bathymetry. Hence, we consider the part of DWBC over the continental slope, indicated by DWBC$_{slope}$ (between 41.5°N and 43.5°N). The DWBC$_{slope}$ has an initial strength of 16.8 Sv (first 50 model years) and reduces to 2.6 Sv (last 50 model years), with the GLORYS12V1 having a time-mean DWBC$_{slope}$ of 23.3 Sv.

Over the first 360 model years, the DWBC$_{slope}$ is weakening by about 1.5 Sv per century using 101-year sliding windows, and the largest trend of 10.3 Sv per century is found between model years 360 and 460. A break regression analysis[63] also confirms different DWBC$_{slope}$ responses before and after model year 360. This is important as the DWBC, contributing to bottom vortex stretching, first reduces in strength and is then followed by the abrupt GS path transition in model year 392. The contribution by surface Ekman pumping velocity is constant as atmospheric winds are prescribed, which implies that the local vorticity balance is primarily influenced by changes in bottom vortex stretching[17]. Note that there are also baroclinic effects, indicated by the different isopycnal slopes (Fig. 4a, b), which could alter topographic steering[64], but these responses are relatively small below 1000 m depths. It is possible that a minimum DWBC$_{slope}$ strength (i.e., threshold) is needed to induce sufficient bottom vortex stretching for a more southward GS path. Exploring such a threshold requires a detailed vorticity analysis at the GS crossover point, which is beyond the scope of this study,

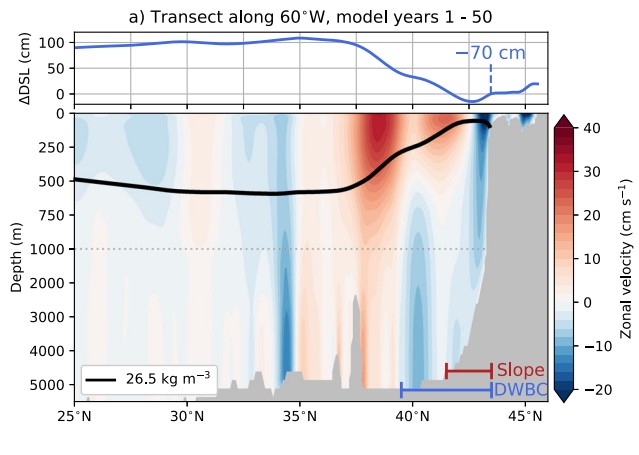

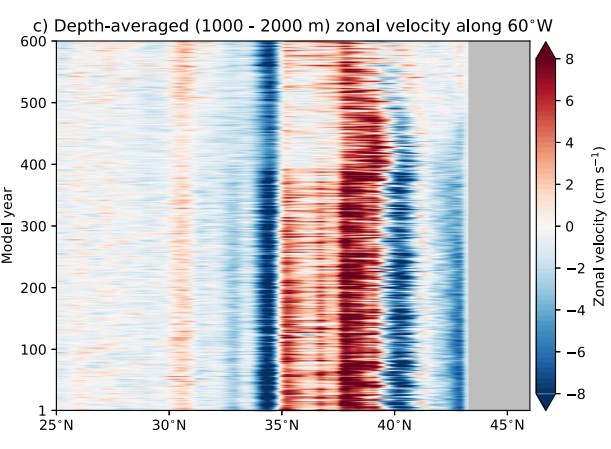

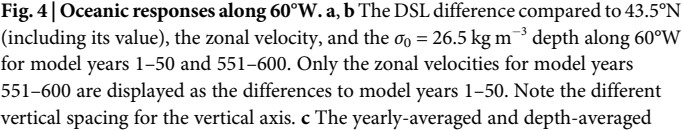

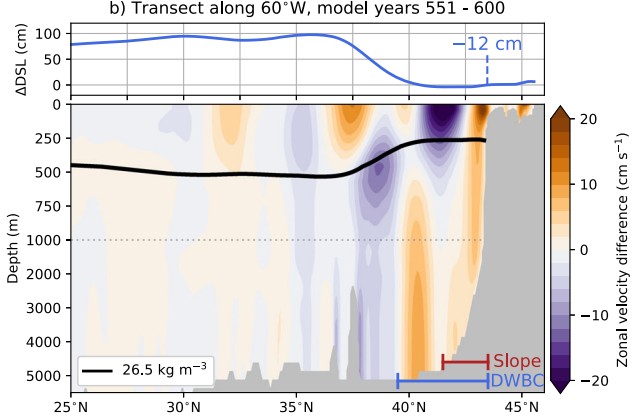

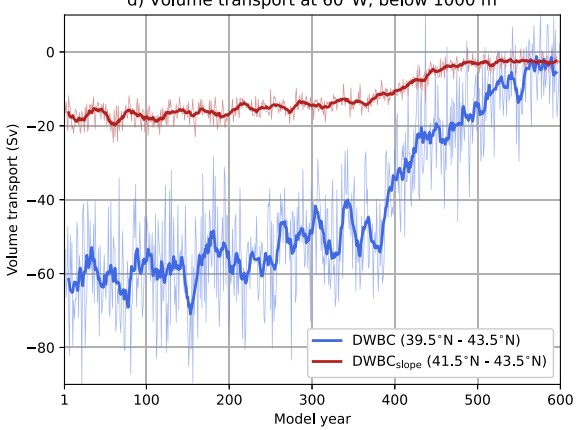

**Fig. 4 | Oceanic responses along 60°W. a**, **b** The DSL difference compared to 43.5°N (including its value), the zonal velocity, and the $\sigma_0 = 26.5$ kg m$^{-3}$ depth along 60°W for model years 1–50 and 551–600. Only the zonal velocities for model years 551–600 are displayed as the differences to model years 1–50. Note the different vertical spacing for the vertical axis. **c** The yearly-averaged and depth-averaged (1000 m to 2000 m) zonal velocity along 60°W. **d** The volume transports below 1000 m for the Deep Western Boundary Current (DWBC, 39.5°N–43.5°N) and the DWBC along the continental slope (DWBC$_{slope}$, 41.5°N–43.5°N). The thin curves are yearly-averaged volume transports, whereas the thick curves are smoothed versions (11-year moving window).

and may aid in interpreting the (25-year) lead-lag timescale between GS path shifts and the onset of the AMOC collapse.

## Warming along the continental slope

A weaker AMOC causes a distinct sea surface temperature (SST) dipole over the North Atlantic Ocean, with cooling over the subpolar gyre and warming north of the GS[14,35]. The POP also shows this SST dipole pattern[50] and we focus here on the warm pool (Fig. 5a). The warming between 75°W and 69°W is caused by the shifting GS as the sharp GS temperature front can reach further north. This is also demonstrated for the local SST at 38°N and 71.5°W (inset in Fig. 5a), which is initially north of the GS and, after the GS shift, south of the GS. This local SST also shows an abrupt increase of about 4 °C within 2 years (model year 392–394), and this local increase is even more striking when considering the upper 250 m temperature with an increase of 6.5 °C over the same two years (Fig. S5a, b).

The higher SSTs south of Nova Scotia and Newfoundland, and in the Gulf of Maine, are not related to the shifting GS, and this can be motivated by considering two effects. First, the GS shifts southward between 69°W and 50°W (Fig. 2c) and this is also the case for the GS north wall (15 °C isotherm at 200 m depth, Fig. 5b). The GS north wall at 60°W shifts southward by 1.3° (145 km) which is a factor of 2 larger than the DSL-GS shift of 0.63° (70 km) at the same longitude. Lower temperatures are found where the GS shifts southward, e.g., the local SST at 37.5°N and 68°W drops by 3.5 °C between the last and first 50 model years. Second, the local temperatures south of Nova Scotia and Newfoundland, and in the Gulf of Maine, reach their equilibrium state after

model year 450 (Fig. S5b). This implies that this region takes longer time to equilibrate compared to 38°N and 71.5°W, where the longer time scales are likely related to the horizontal advection of heat by the (downstream) Labrador Current. Indeed, the largest temperature changes are found over the upper 250 m and between 42.5°N and the continental slope (i.e., 44°N, Fig. 5b), and the Labrador Current is influencing this part of the water column (Fig. 5c, d, see also Figs. 3a and 4a). The local surface heat flux also changes sign, from oceanic heat gain to oceanic heat loss[54], meaning that the warming is not caused by the (prescribed) atmospheric near-surface temperatures.

The (downstream) Labrador Current is advecting relatively cold water masses of subpolar origin along the continental slope. Over the upper 250 m and between 42.5°N and 44°N, the current transports 3 Sv and 45 TW of heat westward and further downstream (Fig. S5c, d). When the AMOC collapses, the Labrador Current ceases to exist (Fig. 3b) and with that the advection of relatively cold water masses, resulting in warming. Accelerated warming has been observed along the continental slope between 2005 and 2017[37], which could suggest that the present-day Labrador Current is weakening. This current also acts as a barrier limiting the influence of ocean eddies close to the continental slope, where the effect of ocean eddies is quantified here by the eddy kinetic energy (Fig. S5e, f). In the absence of the Labrador Current, ocean eddies can easier reach the continental slope, with an increase in the local eddy kinetic energy up to a factor of 7 near 60°W and 43°N. This enhanced eddy activity is also reflected in an increased variance of the zonal heat transport (Fig. S5c), with a factor of 38 difference between the first and last 50 model years.

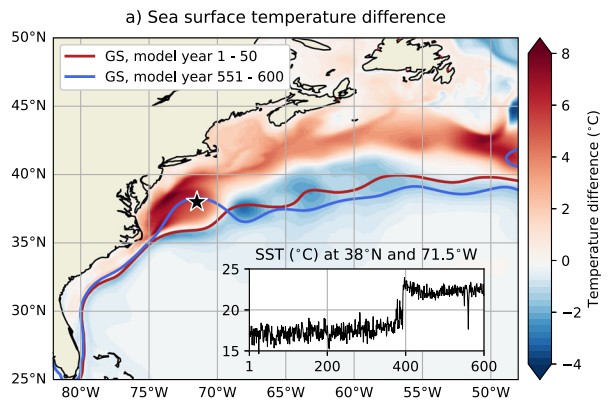

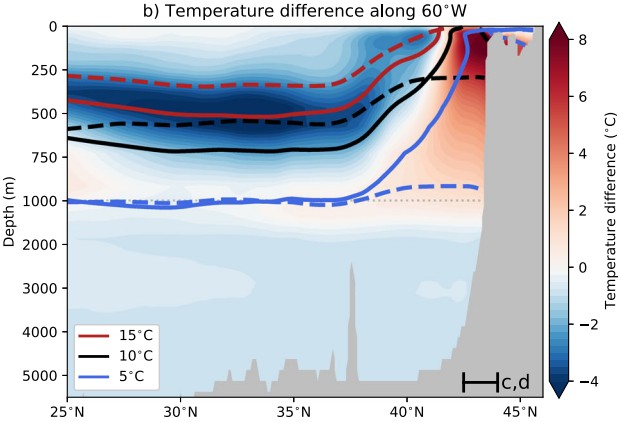

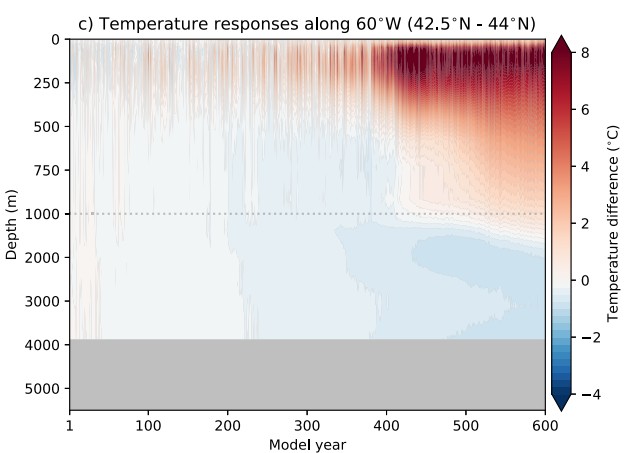

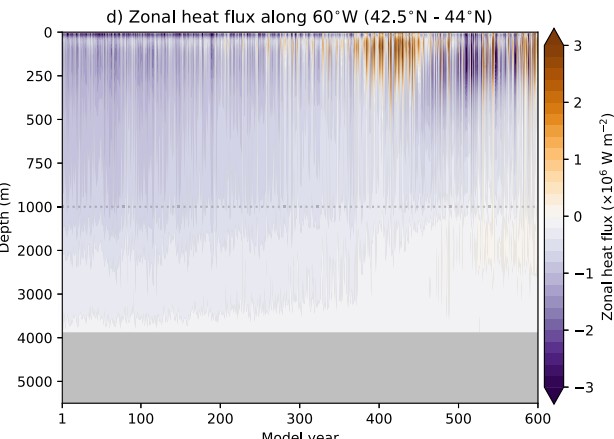

**Fig. 5 | Oceanic temperature responses. a** Sea surface temperature (SST) difference between model years 551–600 and 1–50, including the GS paths. The inset shows the yearly-averaged SST at 38°N and 71.5°W. **b** Temperature responses along 60°W between model years 551–600 and 1–50, including three isotherms for model year 1–50 (solid curves) and model year 551–600 (dashed curves). **c, d** Hovmöller diagrams of the yearly-averaged temperature and zonal heat flux along 60°W and between 42.5°N and 44°N, where the temperature responses are displayed as the difference compared to model year 1–50. Note the different spacing for the vertical axis in (**b–d**).

To summarise, a collapsing AMOC in POP leads to warming along the continental slope. The westernmost part of this warming region, near the GS separation point, is directly influenced by the shifting GS. Further east, the warming is primarily driven by a weakening Labrador Current and increased ocean eddy activity. The results from the POP simulation here nicely isolate the AMOC contribution to this warming, which is challenging in observations or in climate models forced under climate change scenarios[14].

## Shifting Gulf Stream in observations

The POP results show that a northward shifting GS near 71.5°W indicates AMOC weakening and also can also serve as an early warning indicator of the onset of a collapsing AMOC. It is therefore interesting to quantify GS path changes in available observations, where we use satellite altimetry from the AVISO product (1 January 1993 to 1 May 2025). Our outlined DSL procedure for tracking the GS path works well in altimetry and is shown here for 1993 and 2023 (Fig. 6a). Comparing these 2 years demonstrates a northward shifting GS path near the separation point and is in agreement with pervious work[39]. Note that the zero wind-stress curl line can alter the GS path[4,15,65], but has not changed much over the past decades (Fig. 6b).

The largest meridional GS trend of 0.16°N per decade is found at 71.5°W and is significant at $p < 0.05$ (Figs. 6c, d); most trends between 75°W and 69°W are significant at $p < 0.1$. These trends are determined using 32 yearly GS paths (1993–2024), which are obtained from the yearly-averaged

altimetry fields, and the significance of the trend is adjusted for statistical dependency[66]. There is a clear change in the GS path[39], where the GS shifts northward between 75°W and 69°W, and southward between 69°W and 50°W. These GS shifts are qualitatively similar to the POP results (Fig. 2c). The time-mean GS path in POP for model years 360–390 is close to observations, with the GS in POP being only about 0.5°S compared to the observed GS path (blue curve in Fig. 6c). This period is used for comparison as the simulated GS at 71.5°W is closest to observations. The standard deviation in the yearly-averaged GS latitude at 71.5°W is 0.263° (1993–2024, any trend was removed) and is comparable to that in the POP with 0.245°. The GS destabilisation point (inset in Fig. 6a) shifted westward from 1993 to 2014 and shifted eastward again in recent years, consistent with earlier work[11,59].

The GS at 71.5°W is positioned relatively far north in recent years, with a yearly-averaged latitude of 37.8°N in 2022 and 2023, and even reaching 38°N when using the daily-averaged GS path (see inset in Fig. 6d). This latter result is relevant as the GS latitude at 71.5°W in the POP reached that far north prior to the abrupt GS transition (Fig. 2d). The POP also displays GS path trends (at 71.5°W) of at least 0.16°N per decade (using 32-year moving windows) between model years 355 to 386, where this trend window's end is only 6 model years prior to the abrupt GS shift. Note that similar trends are found before model year 300 and are associated with natural multi-decadal variability of Southern Ocean origin[67,68], but for these periods (e.g., model years 10–41), the GS at 71.5°W is positioned further south and is located at 36°N.

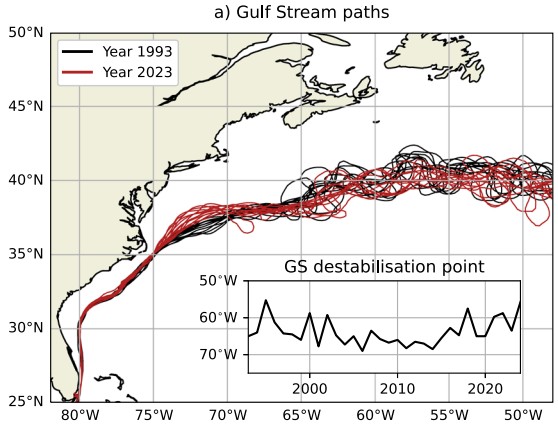

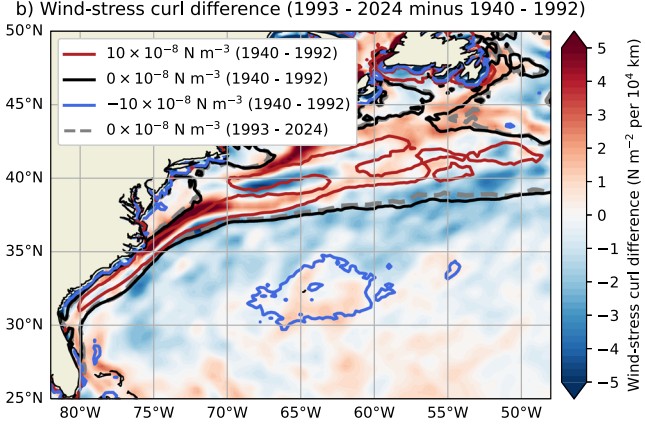

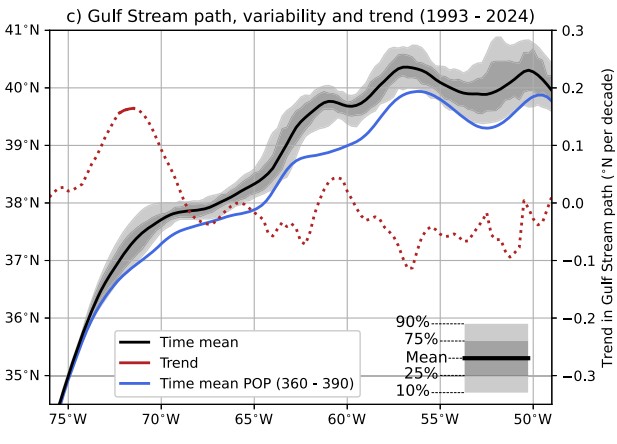

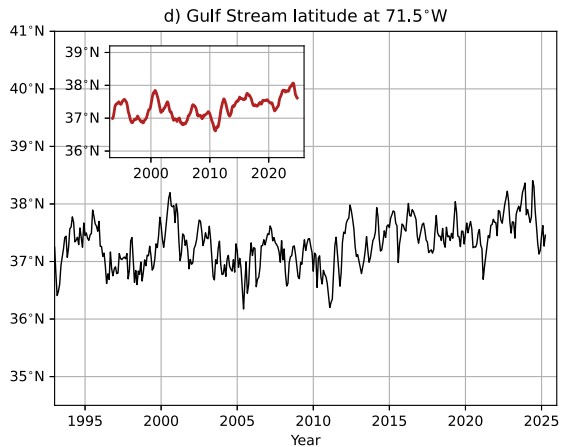

**Fig. 6 | Gulf Stream path in observations. a** The monthly-averaged GS paths from altimetry for the year 1993 (black curves) and the year 2023 (red curves). The insets show the GS destabilisation point. **b** The wind stress curl differences in reanalysis (ERA5) between model years 1993–2024 minus model years 1940–1992, including various isolines of wind stress curl. **c** The GS path for 1993–2024, where first the time-mean altimetry field is obtained and then the GS path is determined. The shading represents the GS path variability using 32 different years, where first the

yearly-averaged altimetry field is obtained and then the GS path is determined. The trend is determined over these 32 yearly GS paths, where the solid curve indicates a significant trend ($p < 0.05$,[66]). The blue curve shows the POP time mean for model years 360–390. **d** The monthly-averaged GS latitude at 71.5°W from altimetry. The insets show the daily-averaged GS latitude, which is smoothed through a 365-day moving average.

More evidence of a shifting GS comes from the GS north wall (i.e., 15°C isotherm at 200 m depth), which has been tracked from in situ seawater temperature profile data since 1965[9]. We use the observational product EN4.2.2 to quantify the GS north wall, which assimilates all available temperature (and salinity) profiles since 1900[69]. Note that data coverage is sparse before 1965 (red curve in Fig. S6a), but the GS north wall at 71.5°W is significantly ($p < 0.01$) shifting northward since 1965 (Fig. S6). The GS north wall is shifting northward over all the presented longitudes (Fig. S6b), whereas altimetry indicates both northward (75°W–69°W) and southward (69°W–50°W) changes in the GS path (Fig. 6c). This difference could either be related to different GS definitions and/or to climate change. The former is not very likely as the POP shows very similar responses (compare Figs. 2c and S2j), while the latter induces a coherent northward shift of the 15 °C (and other) isotherm(s) under warming, which is also noted in recent work[70]. Nevertheless, both GS definitions can be compared at 71.5°W, where the GS path is relatively confined compared to the strongly meandering downstream region[59] (see also Fig. 6a). At this longitude, the GS north wall and GS path from altimetry (blue curve in Fig. S6a) agree well with a root-mean-square error of 0.18°. The trend here is also 0.16°N per decade for the GS north wall (1993–2024), however, it is not significant (blue dashed line in Fig. S6c). Still, the GS north wall at 71.5°W is positioned relatively far north since 2015 ($p < 0.01$, two-sided z-test) compared to 1970–2010, a period with near stationary conditions (yellow dashed line in Fig. S6c). In conclusion, both the GS path from altimetry and the GS north wall indicate a northward shifting GS at 71.5°W in recent years. This is important as these results would suggest that the present-day AMOC is weakening.

## Discussion

We explored the AMOC-GS connection using a recent quasi-equilibrium simulation with a strongly-eddying version of the POP. In this simulation, the AMOC collapses under a slowly increasing freshwater flux forcing to a state with a weak circulation of 5 Sv (−75%) at 26.5°N[50]. This is a feedback-induced collapse because of the large ratio between the AMOC change and freshwater surface forcing change. The high horizontal resolution of 0.1° in POP is needed to realistically resolve western boundary currents[7,45–48], in particular the GS. Although one would like to have such results for additional ocean and global climate models, the POP is, up to now, the only strongly-eddying model where a feedback-induced AMOC collapse has been simulated.

The GS near the separation point at Cape Hatteras (71.5°W) is slowly shifting northward by 133 km during the first 392 model years, which is followed by an abrupt transition of 219 km within 2 years (model years 392–394). This abrupt shift is accompanied by a local temperature increase of 6.5 °C over the same 2 years. Considering the observed standard deviation of the GS path at 71.5°W from altimetry, we expect that such a shift would be significant in observations. The GS path shifts southward further downstream (69°W–50°W), and these meridional shifts are robust when using the GS north wall. The meridionally shifting GS due to the weakening of the AMOC can be understood from theory, because the DWBC along the continental slope affects the vorticity balance at the GS crossover point[14,15,17,18]. Therefore, a northward shifting GS at 71.5°W provides insights into the tendency of the lower and southward flowing AMOC limb. The GS destabilisation point remains fairly constant during the AMOC weakening phase and cannot be used to detect AMOC weakening. The abrupt GS shift happens 25 years before the onset of the AMOC collapse, which can then serve as an effective early warning signal. Do note that this 25-year timescale is likely dependent on the climate model configuration used, bathymetry that shapes the vorticity balance[17,18], and imposed forcing conditions. For the latter, anthropogenic climate change imposes a different forcing on the AMOC than the freshwater flux forcing used here[25]. Consequently, the timing between the abrupt GS shift and onset of the AMOC collapse is likely different for the real AMOC, and a more rigorous vorticity balance analysis near the GS crossover point is needed to understand what determines this time delay.

Satellite altimetry indicates a significant ($p < 0.05$) northward shifting GS near the separation point at 71.5°W over the past three decades (1993–2024). Together with the decreasing DWBC strength[62], these findings suggest that the AMOC is weakening under present-day climate change. The GS trends in altimetry are now becoming significant when using at least 32 years, and this can be complemented by the GS north wall latitude at 71.5°W, which is also significantly ($p < 0.01$) shifting northward since 1965. Further downstream, the GS path shifts have different responses between altimetry (southward) and subsurface temperatures (northward). This discrepancy is attributed to climate change that shifts isotherms northward, hence it is more useful to represent the GS with variables that are independent of varying background conditions.

Reanalysis products, such as GLORYS12V1, provide more information on the oceanic state and agree very well with available observations and altimetry (Fig. S7). For example, the root-mean-square error in the monthly GS path at 71.5°W is 0.19° (21 km) for the GLORYS12V1. There is a significant ($p < 0.05$) GS trend at 71.5°W of 0.17°N per decade between 1993 and 2024, and the DWBC$_{slope}$ is also significantly ($p < 0.05$) weakening by 2.0 Sv per decade over the same period (Fig. S3e). The AMOC time series in the GLORYS12V1 does not show significant trends. Other reanalysis products have even longer temporal extents, such as the SODA3.15.2 (1980–2024) and ORAS5 (1958–2024), but have larger biases than the GLORYS12V1 (Fig. S7). For example, the root-mean-square errors in the monthly GS path at 71.5°W are 0.70° (78 km, SODA3.15.2) and 1.36° (151 km, ORAS5), the ORAS5 having a GS path near the separation point, which is too far north compared to altimetry. These larger biases in SODA3.15.2 and ORAS5 are likely related to their horizontal resolution of 1/4°, which is too coarse to realistically represent the GS[48,71], and make these products less useful for detecting abrupt GS shifts.

A key strength of the POP simulation is its stand-alone ocean configuration with a prescribed atmospheric state, such that the simulated GS changes can be attributed to a substantially weaker AMOC. However, a weaker AMOC impacts the wider climate system[27,28,49], which may affect the GS path through wind responses[51]. Atmospheric modes of variability also influence the GS path in the observational record[41,42], but this connection cannot be captured in the POP because the winds are prescribed. Simulations with a high-resolution version of the CESM (0.1° ocean and 0.25° atmosphere) under constant pre-industrial greenhouse gas forcing and under a climate change scenario are available[72]. Under pre-industrial conditions, the GS at the separation point remains too far north throughout the entire simulation compared to observations, likely due to a relatively weak DWBC (Figure S8). The climate change simulation was branched at model year 250 of the pre-industrial one, but although the AMOC is weakening, it is not very useful to analyse as the GS at the separation point is not in agreement with observations. The GS differences between POP and CESM are intriguing and could be related to climate feedbacks and/or atmospheric variability that drives the GS into a too northerly path[61,73], but the precise mechanisms are outside the scope of this paper.

There is growing evidence that the present-day AMOC is weakening under climate change[34,35,74], and that the AMOC is on route to tipping[24,49,50]. The presented GS-AMOC analysis here provides additional evidence that the present-day AMOC is indeed weakening and that a rapidly shifting GS path can serve as an early warning indicator for the onset of an AMOC tipping event. The advantage of this GS early warning indicator is that it can be observed in near-real time, while other proposed physics-based AMOC[25,49] or statistical[75] indicators require much longer observations. If the AMOC is indeed on route to tipping under future climate change, we expect an abrupt northward GS shift in the upcoming decades.

## Methods
### The Parallel Ocean Program (POP)
We analyse model output of the global Parallel Ocean Program (POP, version 2,[76]) in the nominal 0.1° horizontal resolution configuration with 42 non-equidistant vertical layers and including partial bottom cells[77]. This high-resolution POP allows to explicitly resolve mesoscale ocean

processes[45]. The POP is volume-conserving due to the Boussinesq approximation, and the globally-averaged dynamic sea level is close to zero; the residual was uniformly removed from the dynamic sea-level fields.

The atmospheric forcing is seasonally repeating using observed atmospheric temperatures, river run-off fields, and bulk formula, which are derived from the Coordinated Ocean Reference Experiment (CORE) forcing dataset[77,78]. The POP was first ran into an equilibrium by integrating for 300 years[67]. Next, a freshwater flux forcing ($F_H$) over the North Atlantic Ocean (20°N–50°N) was imposed and was compensated elsewhere (at the ocean surface) to conserve ocean salinity. The freshwater flux forcing increases at a slow rate of $3 \times 10^{-4}$ Sv yr$^{-1}$ up to a maximum forcing of $F_H = 0.18$ Sv in model year 600. We report our POP results in model years, but these can easily be converted to units in freshwater flux forcing strength using the rate in $F_H$. The model output is analysed at a monthly frequency, and most quantities are subsequently converted to yearly averages. More details on this POP simulation can be found in ref. 50.

### Volume transports

The zonal volume transport (ZVT) along a given meridional section ($y_1$ to $y_2$) and over a part of the water column ($z_1$ to $z_2$) is given by:

$$\text{ZVT}(x) = \int_{z_1}^{z_2} \int_{y_1}^{y_2} u(x, y', z') \, dy' dz', \tag{1}$$

with $u$ the zonal velocity. A similar expression holds for the meridional volume transport (MVT) along a given zonal section ($x_1$ to $x_2$):

$$\text{MVT}(y) = \int_{z_1}^{z_2} \int_{x_1}^{x_2} v(x', y, z') \, dx' dz', \tag{2}$$

with $v$ the meridional velocity. We only use the vertical fraction of a grid cell that intersects with $z_1$ or $z_2$ and contributes to ZVT and MVT. The AMOC strength is defined as the MVT between the western coastline ($x_W$) and eastern coastline ($x_E$) in the Atlantic Ocean. For example, the AMOC strength over the upper 1000 m is then determined as:

$$\text{AMOC}(y) = \int_{-1000}^{0} \int_{x_W}^{x_E} v \, dx' dz' \tag{3}$$

### The Gulf Stream path

The GS path is quantified by tracking a DSL isoline (see main text and Fig. 2) or the 15 °C isotherm at 200 m depth. The GS path is determined using monthly-averaged fields or (multi) yearly-averaged fields, where the latitude of the GS path is interpolated to a regular 0.25° longitude grid. In particular, for the monthly-averaged fields, the GS path may have an S-shaped trajectory. In those cases, the most northerly section of the trajectory is used for the interpolation[59]. For the GS destabilisation point, the variance in the monthly-weighted GS latitude is used. The first longitude (from west to east) where the variance exceeds 0.5(°)$^2$ is marked as the GS destabilisation point[59].

### Data availability

The (processed) model output is available at 10.5281/zenodo.17608412. The reanalysis and observational products can be accessed through: GLORYS12V1, AVISO (10.48670/moi-00145), ERA5, EN4.2.2 (https://www.metoffice.gov.uk/hadobs/en4/download-en4-2-2.html), SODA3.15.2 (http://www.soda.umd.edu), and ORAS5. The high-resolution CESM data is available at https://ihesp.github.io/archive/index.html.

### Code availability

The analysis scripts are also available in the zenodo repository (https://doi.org/10.5281/zenodo.16896229).

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

## Acknowledgements

The model simulation and the analysis of all the model output were conducted on the Dutch National Supercomputer Snellius within NWO-SURF project 2024.13. We thank Michael Kliphuis (IMAU, UU) for performing these simulations. R.M.v.W. and H.A.D. are funded by the European Research Council through the ERC-AdG project TAOC (project 101055096, PI: Dijkstra).

## Author contributions

R.M.v.W. and H.A.D. conceived the idea for this study. R.M.v.W. conducted the analysis and prepared all figures. Both authors were actively involved in the interpretation of the analysis results and the writing process.

## Competing interests

The authors declare no competing interests.
