## [Transparent Peer Review file · Communications Earth & Environment]

Abrupt Gulf Stream Path Changes are a Precursor to an AMOC Collapse

Corresponding Author: Dr René van Westen

Version 0:

Decision Letter:

Dear Dr van Westen,

Your manuscript titled "Abrupt Gulf Stream Path Changes are a Precursor to an AMOC Collapse" has now been seen by 3 reviewers, and we include their comments at the end of this message. They find your work of interest, but some important points are raised. We are interested in the possibility of publishing your study in Communications Earth & Environment, but would like to consider your responses to these concerns and assess a revised manuscript before we make a final decision on publication.

We therefore invite you to revise and resubmit your manuscript, along with a point-by-point response that takes into account the points raised. Please highlight all changes in the manuscript text file.

In particular, we would like to request you to consider the following three suggestions:

- 1) Consider including longer observational records;
- 2) Include more relevant works published before and expand discussions on the existing understandings;
- 3) Include more discussions/analyses on the physical mechanisms behind the Gulf Stream shift

Please submit your point-by-point responses as a separate file, distinct from your cover letter where you can add responses to the Editors' comments that you do not want to be made available to the reviewers. Word files are preferred. We recommend that any figures, tables or graphs that are included in the response to reviewers are also included in the main article or Supplementary Information.

Please use the following link to submit your revised manuscript, point-by-point response to the referees' comments (which should be in a separate document to any cover letter), a tracked-changes version of the manuscript (as a PDF file) and the completed checklist:

Link Redacted

We hope to receive your revised paper within six weeks; please let us know if you aren't able to submit it within this time so that we can discuss how best to proceed. If we don't hear from you, and the revision process takes significantly longer, we may close your file. In this event, we will still be happy to reconsider your paper at a later date, as long as nothing similar has been accepted for publication at Communications Earth & Environment or published elsewhere in the meantime.

Please do not hesitate to contact us if you have any questions or would like to discuss these revisions further. We look forward to seeing the revised manuscript and thank you for the opportunity to review your work.

Best regards,

ChenRui Diao, PhD

Associate Editor,
Communications Earth & Environment
Consulting Editor,
Communications Sustainability

EDITORIAL POLICIES AND FORMATTING

- Behavioural and social science
- Ecological, evolutionary & environmental sciences
- Life sciences

Furthermore, please align your manuscript with our format requirements, which are summarized on the following checklist: <https://www.nature.com/documents/commsj-phys-style-formatting-checklist-article.pdf> Communications Earth & Environment formatting checklist

and also in our style and formatting guide <https://www.nature.com/documents/commsj-phys-style-formatting-guide-accept.pdf> Communications Earth & Environment formatting guide .

*** DATA: Communications Earth & Environment endorses the principles of the Enabling FAIR data project (<http://www.copdess.org/enabling-fair-data-project/>). We ask authors to make the data that support their conclusions available in permanent, publically accessible data repositories. (Please contact the editor if you are unable to make your data available).

All Communications Earth & Environment manuscripts must include a section titled "Data Availability" at the end of the Methods section or main text (if no Methods). More information on this policy, is available at <http://www.nature.com/authors/policies/data/data-availability-statements-data-citations.pdf>.

If a community resource is unavailable, data can be submitted to generalist repositories such as <https://figshare.com/> or <http://datadryad.org/> Dryad Digital Repository. Please provide a unique identifier for the data (for example a DOI or a permanent URL) in the data availability statement, if possible. If the repository does not provide identifiers, we encourage authors to supply the search terms that will return the data. For data that have been obtained from publically available sources, please provide a URL and the specific data product name in the data availability statement. Data with a DOI should be further cited in the methods reference section.

REVIEWER COMMENTS:

Reviewer #1 (Remarks to the Author):

A review of van Westen and Dijkstra "Abrupt Gulf Stream Path Changes are a Precursor to an AMOC Collapse".

This paper investigates, primarily in a high resolution (0.1°) ocean model, the Gulf Stream path as a precursor to an AMOC collapse. The study presents the changes to Gulf Stream path, deep western boundary current, temperature change before and after the collapse of the AMOC.

I really welcome this investigation as it is an important point of investigation. The relationship between the AMOC and the Gulf Stream has been investigated extensively in previous studies but not with such a focus on a collapse of the AMOC.

I like the concept. I have three major points that hold me back from recommending publication at this stage:

1. The recent literature (e.g. from Andres, 2016 to Sanchez Roman et al., 2024 and references therein) on the Gulf Stream path has not focused on shifts but on changes to the destabilisation points. I think the destabilisation needs to be investigated as part of this study.
2. The observational support is very limited. It only investigates the past 30 years since the advent of satellite altimetry (since 1993). Gulf Stream North Wall indices are available since the 1960s and the wider context should be investigated. The timescale differences between model (~500 years) and obs (30 years) is stark!
3. The difficulties in the model accurately simulating the Gulf Stream has not been discussed thoroughly. I would like to see a comparison figure of observations and model simulations of GS path. In addition, the relationship of the GSNW to the AMOC is disputed with some authors finding an inverted relationship (Joyce & Zhang, 2010; Zhang & Vallis, 2007) and some authors finding a positive relationship (Nigam et al., 2018). This uncertainty needs to be discussed.

I also have minor points relating to the writing that should be addressed.

Major comments:

There is a striking destabilisation of GS dynamics in 1987. After this point the NAO drives much of the variations in coastal sea level north of Cape Hatteras. This was initially highlighted by Kenigson et al. 2018 and again by Diabaté et al., 2021 (Fig. 4b). To my mind, this is the best observation of an abrupt change in the Gulf Stream. Does this fit in your picture?

L197. I definitely accept the warming on the slope as robustly linked to AMOC change but I don't think you've shown that it is due to subtropical water reaching farther north. This could simply be a reorganisation of the isotherms or it could be less Labrador sea water coming south (my money would be on the latter). The GS front is still a very sharp barrier here. You've shown that the temperature changes abruptly when the GS front crosses a point (Fig 5a inset) but this is quite obvious. Your shift of the GS path to the east of this point doesn't add up in the same way at all. Warming in the Gulf of Maine literature not widely discussed and this needs to be added.

Why do you focus on 60°W? 60°W is very far west to be looking and the behavior of the GS is opposite to what you discuss closer to the coast.

Why would the Lab Current collapse? Isn't there always going to be an eastern boundary current in the subpolar gyre. What does collapse mean? Fig. 3b doesn't really show the Labrador current.

Minor Comments:

I like the 219 km Northward trend in altimetry. This is a number that I think matches observed excursions of the Gulf Stream.

L47. Quite a body of literature about links between the GS and AMOC that hasn't been explored here.

L75. Can the difference be explained by more water flowing in the Antilles current in the model?

L85. I would argue that DWBC observations are not sparse!

Fig. 1a, c: can you superimpose the observed DSL on the model DSL?

Is POP a Boussinesq model? What are the implications for warming and DSL?

L127. I'm not keen on this line. The relationship between the DWBC and GS path is complex and model dependent. The observations are not consistently showing what is stated.

Fig. 4. What do the numbers -70 and -12 cm mean? I can't relate these to anything on the figures.

Fig. 5 a, c are great.

L271. "If the real GS behaves similar as the POP, it would mean that the GS is close of undergoing an abrupt shift, indirectly implying that the onset of the AMOC collapse would follow in a few decades." This line is reaching big time!

I like the 219 km shift, this is quite realistic, I like the temperature increase.

L303. The bracketed (yet) is very loaded and doesn't have a place in a science paper.

L327. Replace "." with " ".

L127. I don't think 'Interestingly' has a place in a scientific publication. Hopefully it's all interesting. I think this is a weasel word.

References:

Andres, M. "On the recent destabilization of the Gulf Stream path downstream of Cape Hatteras." *Geophysical Research Letters* 43, no. 18 (2016): 9836-9842.

Sánchez-Román, Antonio, Flora Gues, Romain Bourdalle-Badie, Marie-Isabelle Pujol, Ananda Pascual, and Marie Drévillon. "Changes in the Gulf Stream path over the last 3 decades." *State of the Planet* 4 (2024): 1-11.

Kenigson, J. S., Han, W., Rajagopalan, B., and Jasinski, M.: Decadal shift of NAO-linked interannual sea level variability along the US northeast coast, *J. Climate*, 31, 4981–4989, <https://doi.org/10.1175/JCLI-D-17-0403.1>, 2018.

Diabaté, Samuel Tiéfolo, Didier Swingedouw, Joël Jean-Marie Hirschi, Aurélie Duchez, Philip J. Leadbitter, Ivan D. Haigh, and Gerard D. McCarthy. "Western boundary circulation and coastal sea-level variability in Northern Hemisphere oceans." *Ocean Science Discussions* 2021 (2021): 1-34.

Joyce, T.M., Zhang, R.: On the path of the gulf stream and the atlantic meridional overturning circulation. *Journal of Climate* 23(11), 3146–3154 (2010)

Zhang, R., Vallis, G.K.: The role of bottom vortex stretching on the path of the north atlantic western boundary current and on the northern recirculation gyre. *Journal of Physical Oceanography* 37(8), 2053–2080 (2007)

Nigam, S., Ruiz-Barradas, A., & Chafik, L. (2018). Gulf Stream excursions and sectional detachments generate the decadal pulses in the Atlantic multidecadal oscillation. *Journal of Climate*, 31(7), 2853–2870.

Reviewer #2 (Remarks to the Author):

Review of the manuscript "Abrupt Gulf Stream Path Changes are a Precursor to an AMOC Collapse" by van Westen and Dijkstra submitted for publication in *Communications Earth & Environment*

This study explores the relationship between changes in the Gulf Stream (GS) path and a potential collapse of the Atlantic Meridional Overturning Circulation (AMOC), a topic of significant interest in oceanography and climatology. The authors employ a high-resolution stand-alone ocean simulation, where a gradual increase in freshwater forcing leads to the AMOC's collapse. Their focus is on the GS's behavior and its connection to AMOC changes. Their results indicate that as the AMOC weakens, the GS near Cape Hatteras shifts northward, initially gradually and then abruptly. The analysis reveals that at 71.5°W, the GS makes a 219 km northward leap within just two years. The authors suggest this could serve as an early warning of the AMOC reaching a tipping point, as the GS changes occur approximately 25 years before that event. They associate the GS shift with a significant weakening of the Deep Western Boundary Current, changes in bottom-vortex stretching, regional warming, and altered eddy activity along the continental slope. Comparing these model results with observations, both satellite altimetry (1993–2023) and a high-resolution reanalysis (1993–2024) show a statistically significant northward trend in the GS near the separation point. The authors propose that abrupt GS shifts could act as an early-warning indicator of an impending AMOC collapse. They argue that recent observational trends provide indirect evidence of an ongoing AMOC decline.

The manuscript is well-written and has many strong points. However, it has a significant issue in that it completely overlooks similar research on the GS path over the six decades, from 1965 to 2017, which is based on historical oceanographic in situ data. A paper published in *Scientific Reports*, titled "Resilience of the Gulf Stream path on decadal and longer timescales" by Seidov and co-authors (Seidov et al., *Scientific Reports*, 2019, doi:10.1038/s41598-019-48011-9), addresses this topic. They utilized the World Ocean Database and tracked the 15°C isotherm at a depth of 200m as a marker of the GS path. Their findings revealed that the region between 75°W and 50°W was very stable, whereas the GS path east of 50°W was much more variable. Seidov and co-authors discovered that the overall northward shift in the path from 1965 to 2017 was minimal (~0.4° in latitude), while east of 50°W, this shift was significantly larger. They noted that the GS between Cape Hatteras and the Grand Banks is highly resilient, which contradicts the modeling results used by van Westen and Dijkstra, who observed a significant northward shift near 75°W.

Seidov and colleagues also noted that, in contrast, east of 50°W, the GS path shifts significantly, moving more than 2.6° in latitude, which equates to approximately 263 km. Interestingly, while the northward jump figures provided by van Westen and Dijkstra and by Seidov and colleagues are similar, they refer to different locations. The GS path jumps near 75°W contradicts in situ data analysis, which shows the strong resilience of the GS path. Furthermore, Seidov et al. connected the GS resilience and variability to the wind-stress curl, drawing parallels to the significance of the wind stress curl in GS path

dynamics (specifically the position of the zero line of the wind stress curl) as discussed in the reviewed manuscript. Given the similarities and differences between the GS path dynamics modeled and analyzed in the publication by Seidov and co-authors and those in the submitted manuscript, it is essential to discuss these distinctions and/or commonalities in detail. This may require substantial restructuring of the text and could necessitate major revisions.

In conclusion, without a direct and thorough comparison between the model and in situ data reconstruction of the GS path, it is challenging to ascertain the merits of the submitted manuscript. The authors should compare their results with the paper by Seidov et al., 2019, and explain the significant deviations. Such explanations necessitate major revisions. Aside from this, the manuscript holds great potential for publication and could significantly contribute to the AMOC research pool. If the authors agree to draw parallels with previously published analyses and successfully explain the differences, the paper may become suitable for publication in *Communications Earth & Environment*.

Other remarks

Lines 79-82: The statements about AMOC and FC strength "... decreasing to xx Sv over the last 50 model years" is contradict the referenced Fig.1 c), where it is shown that mentioned values of 5 and 20 Sv exists over last ~100 model years

Figure 1: I believe that showing a volume transport curves on b) and c) usage of the 5 (or 10)-years-averaged instead of yearly-averaged values will reduce the noise and improve the presentation of the model results (see fig. 2 d) for example).

Figure 1 caption: "Note the non-linear horizontal range and vertical range on the axes" – this statement a bit confusing, since horizontal range on horizontal axes in a) and c) (longitude) looks linear. – Please check and correct

Lines 87 – 89: "the DWBC has an initial strength of 37.9 Sv (first 50 model years), which reduces to 16.5 Sv (last 50 model years). The AMOC strength below 1,000 m depth starts at 21 Sv and declines to 5 Sv. The POP results (first 50 model years)" - similar to remark to lines 79-82 above – Fig 1d) shows that initial strength of 37.9 Sv exists for first 300 model years (not 50), and reduced to 16.5 Sv for last 100 years (not 50). – Please take a look and correct.

Line 142-144: "Another notable difference is the 143 almost complete cessation of the Labrador Current south of Newfoundland and Nova Scotia." - Would be useful to compare with the in-situ observations (see for example Seidov et al. *Limn & Oceanog.* 2021; doi: 10.1002/lno.11892)

Lines 180-181: first 200 model years, not 50 and last 150 model years, not 50. Please take a look and correct.

Fig. 4 d) – similar remark as to Fig 1 – using 10yr-avg will improve the presentation

Line 292: "Satellite altimetry indicates a significant ($p < 0.1$) and northward shifting GS near" – consider remove 'and'

Line 327: "There is growing evidence that the present-day AMOC is weakening under climate" – consider remove '.'

Reviewer #3 (Remarks to the Author):

The authors explored the relationship between the AMOC and the GS using a freshwater hosing experiment with a high-resolution POP. They found that a weaker AMOC progressively displaces the GS path near Cape Hatteras northward, which features an abrupt northward shift within 2 years. This rapid shift occurs 25 years before the simulated AMOC tipping. They suggested that abrupt GS shifts can provide as a near-real time, observable early warning indicator for the commencement of an AMOC tipping event.

This is an interesting study. The paragraphs are well organized, and the figures are well prepared. I appreciate the high-resolution POP freshwater hosing experiment as well as the comprehensive analysis on the experiment output. Thereupon, I feel that the manuscript could be acceptable after proper revisions.

My main comments are on the physical mechanisms behind the Gulf Stream shift. Although the authors well describe the phenomena, they simply referenced previous studies to refer to the mechanism of bottom vortex stretching for changes in the Gulf Stream and DWBC. I respect the statement "Exploring such a threshold requires a detailed vorticity analysis at the GS crossover point, which is beyond the scope of this study.", but I am wondering how understand the Gulf Stream change physically. For example, in the ocean-alone experiment, as per the "Methods" part and ref. 39 "The HR-POP is forced under a prescribed atmospheric state (near-surface atmospheric temperatures and bulk formula) with a repeating seasonal cycle and observed river run-off fields", I guess the surface wind stress is prescribed and will not change during the freshwater hosing. That means the surface wind stress curl will not alter in the vorticity equation, e.g., that in ref. 17, Zhang and Valls (2007)? Shall we understand this equation from a barotropic perspective, or shall we (also) consider the question from a baroclinic perspective, as the freshwater input can alter ocean stratification and hence affect the topography steering (e.g., Marshall 1995)? In another word, this could be somehow helpful to understand the relationship between the Gulf Stream and the AMOC, since the Gulf Stream was considered to include both wind-driven (but fixed wind in the experiment?) and thermohaline components. Additionally, I still would like to highlight the importance of topography, as the bottom pressure torque (BPT) appears as a dominant term in the vorticity balance of the AMOC, representing a key dynamical link between the overturning and gyre streamfunctions in the North Atlantic (Yeager 2015).

Marshall, D., 1995. Influence of topography on the large-scale ocean circulation. *Journal of physical oceanography*, 25, 1622-1635.

Yeager, S., 2015. Topographic coupling of the Atlantic overturning and gyre circulations. *Journal of Physical Oceanography*, 45, 1258-1284.

As related, the authors may want to notice the effect of surface winds as their winds seem to be fixed in the freshwater

hosing experiment in ocean-alone model. In a fully coupled climate model, a weakened AMOC can generate anticyclonic surface wind anomalies over the subpolar North Atlantic to decelerate the subpolar gyre circulation, together with a weaker northward Gulf Stream (Mimi and Liu 2024). Atmospheric wind response to AMOC change, which further influences the Gulf Stream, needs to be discussed from the perspective of ocean-atmosphere coupling.

Mimi, M.S. and Liu, W., 2024. Atlantic Meridional Overturning Circulation slowdown modulates wind-driven circulations in a warmer climate. *Communications Earth & Environment*, 5, 727.

Moreover, the authors mentioned in Lines 124-126, Page 4 that "What is most relevant is that this GS path transition occurs approximately 25 years prior to the onset of the AMOC tipping event. The different GS paths may be related to different GS path equilibria". Can the authors elaborate on the cause of the timescale, in particular, why it should be 25 years?

Besides, since the high-resolution POP is used, I am wondering whether the well-simulated oceanic mesoscale eddies will allow for a better representation of eddy momentum fluxes (ref. 7, Greatbatch et al. 2010) and eddy-mean flow interactions (Kang and Curchitser 2015) in the Gulf Stream Region.

Kang, D. and Curchitser, E.N., 2015. Energetics of eddy-mean flow interactions in the Gulf Stream region. *Journal of Physical Oceanography*, 45, 1103-1120.

** Visit Nature Portfolio's author and referees' website at www.nature.com/authors for information about policies, services and author benefits**

Communications Earth & Environment is committed to improving transparency in authorship. As part of our efforts in this direction, we are now requesting that all authors identified as 'corresponding author' create and link their Open Researcher and Contributor Identifier (ORCID) with their account on the Manuscript Tracking System prior to acceptance. ORCID helps the scientific community achieve unambiguous attribution of all scholarly contributions. You can create and link your ORCID from the home page of the Manuscript Tracking System by clicking on 'Modify my Springer Nature account' and following the instructions in the link below. Please also inform all co-authors that they can add their ORCIDs to their accounts and that they must do so prior to acceptance.

If you experience problems in linking your ORCID, please contact the Platform Support Helpdesk.

Version 1:

Decision Letter:

Dear Dr van Westen,

Your manuscript titled "Abrupt Gulf Stream Path Changes are a Precursor to an AMOC Collapse" has now been seen by our reviewers, whose comments appear below. In light of their advice we are delighted to say that we are happy, in principle, to publish a suitably revised version in *Communications Earth & Environment*.

We therefore invite you to revise your paper one last time to address the remaining concerns of our reviewers. At the same time we ask that you edit your manuscript to comply with our format requirements and to maximise the accessibility and therefore the impact of your work.

EDITORIAL REQUESTS:

*****Please take care to match our formatting and policy requirements. We will check revised manuscript and return manuscripts that do not comply. Such requests will lead to delays. *****

SUBMISSION INFORMATION:

OPEN ACCESS:

Communications Earth & Environment is a fully open access journal. Articles are made freely accessible on publication. For further information about article processing charges, open access funding, and advice and support from Nature Portfolio, please visit <https://www.nature.com/commsenv/open-access>

Link Redacted

Best regards,

ChenRui Diao, PhD

Associate Editor,
Communications Earth & Environment
Consulting Editor,
Communications Sustainability

REVIEWERS' COMMENTS:

Reviewer #1 (Remarks to the Author):

2nd review of van Westen and Dijkstra

I really like this paper, it presents a convincing study of Gulf Stream (GS) shifts in response to AMOC collapse. The inclusion of observational data is very plausible. I think the extended observational GS north wall strengthens the analysis. I was going to suggest to use the Taylor GS index rather than Joyce to avoid the warming problem but it looks like that isn't available any longer. I am happy that the authors have addressed my comments and can recommend the paper for publication.

Minor comments:

l37, suggest replacing 'interesting' with a more quantitative word

l54, suggest diluting the claim or specifying that all the studies cited (14, 34, 35) use SST as an AMOC proxy. Adding "... reconstructions *based on SST data* suggest..."

There are outstanding formatting issues in the references such as incorrect capitalisation (LaTeX issue!)

Taylor, A., & Stephens, J. (1980). Latitudinal displacements of the Gulf-Stream (1966 to 1977) and their relation to changes in temperature and zooplankton abundance in the NE Atlantic. *Oceanologica Acta*, 3(2), 145–149.

Reviewer #2 (Remarks to the Author):

I believe all relevant remarks have been addressed and this MS can be published.
Thank you

Reviewer #3 (Remarks to the Author):

I am generally fine with the authors' reply and revision, and thus recommend acceptance.

** Visit Nature Portfolio's author and referees' website at www.nature.com/authors for information about policies, services and author benefits**

MS-No.: COMMSENV-25-4192-T

Version: Revision

Title: Abrupt Gulf Stream Path Changes are a Precursor to an AMOC Collapse

Author(s): René M. van Westen and Henk A. Dijkstra

Point-by-point reply to reviewers

November 14, 2025

We thank the reviewers for their careful reading and for the useful comments on the revised manuscript. Please find our specific responses listed below. In the meantime, also the observational and reanalysis products were extended. Our results and text are updated accordingly using the latest available data records. Most importantly, the GS path at 71.5°W is now significantly ($p < 0.05$) shifting northward between 1993 to 2024.

Reviewer #1

A review of van Westen and Dijkstra “Abrupt Gulf Stream Path Changes are a Precursor to an AMOC Collapse”.

This paper investigates, primarily in a high resolution (0.1°) ocean model, the Gulf Stream path as a precursor to an AMOC collapse. The study presents the changes to Gulf Stream path, deep western boundary current, temperature change before and after the collapse of the AMOC.

I really welcome this investigation as it is an important point of investigation. The relationship between the AMOC and the Gulf Stream has been investigated extensively in previous studies but not with such a focus on a collapse of the AMOC.

I like the concept. I have three major points that hold me back from recommending publication at this stage:

1. *The recent literature (e.g. from Andres, 2016 to Sanchez Roman et al., 2024 and references therein) on the Gulf Stream path has not focused on shifts but on changes to the destabilisation points. I think the destabilisation needs to be investigated as part of this study.*

We have determined the GS destabilisation point in POP following the procedure outlined in Andres (2016). This point remains fairly constant during the first 392 model years and thereafter shifts 2.84°W . This means that the GS destabilisation point is useful for detecting the GS shift, but it is fairly stable under the AMOC weakening over the first 392 model years. The GS destabilisation point analysis is included in the revised manuscript for POP (revised Figure S2k), observations (revised Figure 6a), and reanalysis products (revised Figure S7). The results are discussed in the revised text.

- 2. The observational support is very limited. It only investigates the past 30 years since the advent of satellite altimetry (since 1993). Gulf Stream North Wall indices are available since the 1960s and the wider context should be investigated. The timescale differences between model (500 years) and obs (30 years) is stark!*

We have included an analysis on the GS north wall in the observational product EN4.2.2 (revised Figure S6). The GS north wall at 71.5°W significantly ($p < 0.01$) shifts northward since 1965, confirming the GS path changes from altimetry (1993 – 2024). These findings are presented in the revised section ‘Shifting Gulf Stream in Observations’.

- 3. The difficulties in the model accurately simulating the Gulf Stream has not been discussed thoroughly. I would like to see a comparison figure of observations and model simulations of GS path. In addition, the relationship of the GSNW to the AMOC is disputed with some authors finding an inverted relationship (Joyce & Zhang, 2010; Zhang & Vallis, 2007) and some authors finding a positive relationship (Nigam et al., 2018). This uncertainty needs to be discussed.*

Revised Figure 6c now includes the time-mean GS path for the POP (model years 360 – 390) for direct comparison with observations. The simulated GS path reasonably agrees with the observed GS path, with the simulated GS path about 0.5° south of observations. The GS destabilisation point is located at 66.3°W , which agrees well with that of observations; this is mentioned in the revision.

Revised Figure S2j now shows the time-mean changes in the GS north wall, with meridional shifts that are very similar to the GS path from DSL fields. In observations, however, the GS north wall shifts northward ($75^\circ\text{W} - 50^\circ\text{W}$) while the GS path from altimetry shifts northward

(75°W – 69°W) and southward (69°W – 50°W). This different behavior is attributed to climate change, which induces a coherent northward shift of the 15°C (and other) isotherm(s). This is further discussed in the revised manuscript.

I also have minor points relating to the writing that should be addressed.

Major comments

1. *There is a striking destabilisation of GS dynamics in 1987. After this point the NAO drives much of the variations in coastal sea level north of Cape Hatteras. This was initially highlighted by Kenigson et al. 2018 and again by Diabaté et al., 2021 (Fig. 4b). To my mind, this is the best observation of an abrupt change in the Gulf Stream. Does this fit in your picture?*

The POP has prescribed atmospheric winds and hence this connection can't be explored. We have commented on this point in the revised discussion.

2. *L197. I definitely accept the warming on the slope as robustly linked to AMOC change but I don't think you've shown that it is due to subtropical water reaching farther north. This could simply be a reorganisation of the isotherms or it could be less Labrador sea water coming south (my money would be on the latter). The GS front is still a very sharp barrier here. You've shown that the temperature changes abruptly when the GS front crosses a point (Fig 5a inset) but this is quite obvious. Your shift of the GS path to the east of this point doesn't add up in the same way at all. Warming in the Gulf of Maine literature not widely discussed and this needs to be added.*

The reviewer is correct that we didn't show that subtropical water masses reach farther north. A more appropriate interpretation is that the sharp GS temperature front shifts northward between 75°W and 70°W. We have rephrased this sentence.

As was already argued in the main text (lines 204 – 218 and Figure S5), the warming over the Gulf of Maine, and the regions south of Nova Scotia and Newfoundland, are related to the cessation of the (downstream)

Labrador Current. The surface heat flux does not contribute to this warming; this was added in the revision.

3. *Why do you focus on 60° W? 60° W is very far west to be looking and the behavior of the GS is opposite to what you discuss closer to the coast.*

As was mentioned in the main text (lines 154 – 155), the results are not sensitive when using a different longitude (see also Figure S4). But we do agree with the reviewer that a motivation for the 60°W location was missing. This motivation is now included in the revision.

4. *Why would the Lab Current collapse? Isn't there always going to be an eastern boundary current in the subpolar gyre. What does collapse mean? Fig. 3b doesn't really show the Labrador current.*

The reviewer is correct, the upstream Labrador Current (in the Labrador Sea) is not collapsing (see Zenodo repository). The downstream Labrador Current, flowing south of Newfoundland and Nova Scotia, does collapse. We were specifically referring to this downstream branch in the manuscript, which was somewhat confusing. We have clarified this in the revision.

Minor comments

1. *I like the 219 km Northward trend in altimetry. This is a number that I think matches observed excursions of the Gulf Stream.*

Thank you.

2. *L47. Quite a body of literature about links between the GS and AMOC that hasn't been explored here.*

We added a new paragraph in the introduction that discusses the links between the AMOC and GS.

3. *L75. Can the difference be explained by more water flowing in the Antilles current in the model?*

This could be an explanation, but a complete analysis on Florida Current strength biases distracts from the overall aim of the manuscript. No changes in manuscript.

4. *L85. I would argue that DWBC observations are not sparse!*

Agreed, sentence was rewritten.

5. *Fig. 1a, c: can you superimpose the observed DSL on the model DSL?*

The differences between observed DSL and modelled DSL are relatively small and may confuse the reader as the DSL subpanel is relatively small. Therefore, we kept the figure as is and refer to Figure S1 for comparison.

6. *Is POP a Boussinesq model? What are the implications for warming and DSL?*

The POP is indeed a Boussinesq model and is volume conserving. This means that the globally-averaged DSL is close to zero and the residual was uniformly removed from the DSL fields. There are no implications for warming, however, because steric effects need to be determined by post-processing the model output (see <https://doi.org/10.5194/egusphere-2025-5102>).

In the revised Methods, we mention that the POP is a Boussinesq model and how DSL was processed.

7. *L127. I'm not keen on this line. The relationship between the DWBC and GS path is complex and model dependent. The observations are not consistently showing what is stated.*

Agreed. The opening sentence of the Section ‘Changes in the Deep Western Boundary Current’ is modified, acknowledging the more complicated relationship between DWBC and GS in observations than in the process based studies cited.

8. *Fig. 4. What do the numbers -70 and -12 cm mean? I can't relate these to anything on the figures.*

These are the DSLs at 43.5°N, as was mentioned in the caption, and is comparable to Figure 1. No changes in the manuscript.

9. *Fig. 5 a, c are great.*

Thank you.

10. *L271. “If the real GS behaves similar as the POP, it would mean that the GS is close of undergoing an abrupt shift, indirectly implying that the onset of the AMOC collapse would follow in a few decades.” This line is reaching big time!*

We agree with the reviewer on this. Furthermore, we do not know whether the real AMOC and GS have a similar lead-lag relation as in our idealised hosing set-up. We have removed this line and extended the discussion on this 25-year timescale in the revision.

11. *I like the 219 km shift, this is quite realistic, I like the temperature increase.*

Thank you.

12. *L303. The bracketed (yet) is very loaded and doesn't have a place in a science paper.*

Agreed, we have removed ‘(yet)’ here.

13. *L327. Replace “.” with “ “.*

Corrected.

14. *L127. I don't think ‘Interestingly’ has a place in a scientific publication. Hopefully it's all interesting. I think this is a weasel word.*

Suggestion followed.

Reviewer #2

Review of the manuscript “Abrupt Gulf Stream Path Changes are a Precursor to an AMOC Collapse” by van Westen and Dijkstra submitted for publication in Communications Earth & Environment

This study explores the relationship between changes in the Gulf Stream (GS) path and a potential collapse of the Atlantic Meridional Overturning Circulation (AMOC), a topic of significant interest in oceanography and climatology. The authors employ a high-resolution stand-alone ocean simulation, where a gradual increase in freshwater forcing leads to the AMOC’s collapse. Their focus is on the GS’s behavior and its connection to AMOC changes. Their results indicate that as the AMOC weakens, the GS near Cape Hatteras shifts northward, initially gradually and then abruptly. The analysis reveals that at 71.5°W, the GS makes a 219 km northward leap within just two years. The authors suggest this could serve as an early warning of the AMOC reaching a tipping point, as the GS changes occur approximately 25 years before that event. They associate the GS shift with a significant weakening of the Deep Western Boundary Current, changes in bottom-vortex stretching, regional warming, and altered eddy activity along the continental slope. Comparing these model results with observations, both satellite altimetry (1993–2023) and a high-resolution reanalysis (1993–2024) show a statistically significant northward trend in the GS near the separation point. The authors propose that abrupt GS shifts could act as an early-warning indicator of an impending AMOC collapse. They argue that recent observational trends provide indirect evidence of an ongoing AMOC decline.

The manuscript is well-written and has many strong points. However, it has a significant issue in that it completely overlooks similar research on the GS path over the six decades, from 1965 to 2017, which is based on historical oceanographic in situ data. A paper published in Scientific Reports, titled “Resilience of the Gulf Stream path on decadal and longer timescales” by Seidov and co-authors (Seidov et al., Scientific Reports, 2019, doi:10.1038/s41598-019-48011-9), addresses this topic. They utilized the World Ocean Database and tracked the 15°C isotherm at a depth of 200m as a marker of the GS path. Their findings revealed that the region between 75°W and 50°W was very stable, whereas the GS path east of 50°W was much more variable. Seidov and co-authors discovered that the overall northward shift in the path from 1965

to 2017 was minimal (0.4° in latitude), while east of 50°W , this shift was significantly larger. They noted that the GS between Cape Hatteras and the Grand Banks is highly resilient, which contradicts the modeling results used by van Westen and Dijkstra, who observed a significant northward shift near 75°W .

We have quantified the GS north wall using observational product EN4.2.2 (revised Figure S6), which is available since 1900. Data coverage is sparse before 1965, consistent with Seidov et al. (2019), and there is a significant ($p < 0.01$) trend in the GS north wall at 71.5°W between 1965 to 2024. The GS north wall is indeed quite stationary between 1970 to 2010. These findings are discussed in the revised section ‘Shifting Gulf Stream in Observations’.

Seidov and colleagues also noted that, in contrast, east of 50°W , the GS path shifts significantly, moving more than 2.6° in latitude, which equates to approximately 263 km. Interestingly, while the northward jump figures provided by van Westen and Dijkstra and by Seidov and colleagues are similar, they refer to different locations. The GS path jumps near 75°W contradicts in situ data analysis, which shows the strong resilience of the GS path. Furthermore, Seidov et al. connected the GS resilience and variability to the wind-stress curl, drawing parallels to the significance of the wind stress curl in GS path dynamics (specifically the position of the zero line of the wind stress curl) as discussed in the reviewed manuscript. Given the similarities and differences between the GS path dynamics modeled and analyzed in the publication by Seidov and co-authors and those in the submitted manuscript, it is essential to discuss these distinctions and/or commonalities in detail. This may require substantial restructuring of the text and could necessitate major revisions.

Our analysis focusses on the GS path between 75°W and 50°W . Larger GS changes can be expected further downstream, as the GS transitions to a strongly meandering jet. The latter has now been quantified in the revised manuscript (following suggestion by Reviewer #1). Additional analysis of the GS north wall (up to 2024) demonstrates that the GS is shifting north in recent years (revised Figure S6). This does not contradict the results by Seidov et al. (2019), as they analysed the GS north wall up to 2017. Analysing GS path changes east of 50°W is beyond the scope of the manuscript.

In conclusion, without a direct and thorough comparison between the model and in situ data reconstruction of the GS path, it is challenging to ascertain the merits of the submitted manuscript. The authors should compare their results with the paper by Seidov et al., 2019, and explain the significant deviations. Such explanations necessitate major revisions. Aside from this, the manuscript holds great potential for publication and could significantly contribute to the AMOC research pool. If the authors agree to draw parallels with previously published analyses and successfully explain the differences, the paper may become suitable for publication in Communications Earth & Environment.

We also present the time-mean changes in the GS north wall for the POP in revised Figure S2j. It turns out that these meridional shifts are very similar to the GS path from DSL fields. In observations, however, the GS north wall shifts northward ($75^{\circ}\text{W} - 50^{\circ}\text{W}$) while the GS path from altimetry shifts northward ($75^{\circ}\text{W} - 69^{\circ}\text{W}$) and southward ($69^{\circ}\text{W} - 50^{\circ}\text{W}$). This different behavior is attributed to climate change, that causes a coherent northward shift of the 15°C (and other) isotherm(s). The revised manuscript now discusses in greater detail the changes in the GS north wall.

Other remarks

1. *Lines 79-82: The statements about AMOC and FC strength "... decreasing to xx Sv over the last 50 model years" is contradict the referenced Fig.1 c), where it is shown that mentioned values of 5 and 20 Sv exists over last 100 model years*

The reviewer is correct here. However, we only compare the first and last 50 model years as most time series and the AMOC are still equilibrating between model year 500 to 550; this is now clarified in the revision. Using different windows may be confusing and therefore we primarily focus on the first and last 50 model years in the main text.

2. *Figure 1: I believe that showing a volume transport curves on b) and c) usage of the 5 (or 10)-years-averaged instead of yearly-averaged values will reduce the noise and improve the presentation of the model results (see fig. 2 d) for example).*

This is an excellent suggestion! We now show in revised Figures 1b,d the 11-year moving average as thick curves and the thin lines are the yearly-averaged values.

3. *Figure 1 caption: “Note the non-linear horizontal range and vertical range on the axes” – this statement a bit confusing, since horizontal range on horizontal axes in a) and c) (longitude) looks linear. – Please check and correct*

This was indeed confusing because we used a different (linear) spacing. It has been clarified in the revised Figure captions.

4. *Lines 87 – 89: “the DWBC has an initial strength of 37.9 Sv (first 50 model years), which reduces to 16.5 Sv (last 50 model years). The AMOC strength below 1,000 m depth starts at 21 Sv and declines to 5 Sv. The POP results (first 50 model years)” - similar to remark to lines 79-82 above – Fig 1d) shows that initial strength of 37.9 Sv exists for first 300 model years (not 50), and reduced to 16.5 Sv for last 100 years (not 50). – Please take a look and correct.*

The reviewer is also correct here, but for consistency, we keep comparing the first and last 50 model years. No changes in the manuscript.

5. *Line 142-144: “Another notable difference is the almost complete cessation of the Labrador Current south of Newfoundland and Nova Scotia.” - Would be useful to compare with the in-situ observations (see for example Seidov et al. Limn & Oceanog. 2021*

Seidov et al. (2021) report accelerated warming along the continental slope, which could be related to a weaker Labrador Current. This is now mentioned in the revised paper.

6. *Lines 180-181: first 200 model years, not 50 and last 150 model years, not 50. Please take a look and correct. Fig. 4 d) – similar remark as to Fig 1 – using 10yr-avg will improve the presentation*

For consistency, we compare the first and last 50 model years. We followed the reviewer’s suggestion and the time series are now also smoothed through a 11-year moving average.

7. *Line 292: “Satellite altimetry indicates a significant ($p < 0.1$) and northward shifting GS near” – consider remove ‘and’*

Suggestion followed.

8. *Line 327: “There is growing evidence that.the present-day AMOC is weakening under climate” – consider remove ‘.’*

Corrected.

Reviewer #3

The authors explored the relationship between the AMOC and the GS using a freshwater hosing experiment with a high-resolution POP. They found that a weaker AMOC progressively displaces the GS path near Cape Hatteras northward, which features an abrupt northward shift within 2 years. This rapid shift occurs 25 years before the simulated AMOC tipping. They suggested that abrupt GS shifts can provide as a near-real time, observable early warning indicator for the commencement of an AMOC tipping event.

This is an interesting study. The paragraphs are well organized, and the figures are well prepared. I appreciate the high-resolution POP freshwater hosing experiment as well as the comprehensive analysis on the experiment output. Thereupon, I feel that the manuscript could be acceptable after proper revisions.

My main comments are on the physical mechanisms behind the Gulf Stream shift. Although the authors well describe the phenomena, they simply referenced previous studies to refer to the mechanism of bottom vortex stretching for changes in the Gulf Stream and DWBC. I respect the statement “Exploring such a threshold requires a detailed vorticity analysis at the GS crossover point, which is beyond the scope of this study.”, but I am wondering how understand the Gulf Stream change physically. For example, in the ocean-alone experiment, as per the “Methods” part and ref. 39 “The HR-POP is forced under a prescribed atmospheric state (near-surface atmospheric temperatures and bulk formula) with a repeating seasonal cycle and observed river run-off fields”, I guess the surface wind stress is prescribed and will not change during the freshwater hosing. That means the surface wind stress curl will not alter in the vorticity equation, e.g., that in ref. 17, Zhang and Valls (2007)?

Correct, the surface wind stress is prescribed and is seasonally varying, we have emphasised this in the revised Introduction. Indeed, the contribution by the surface wind stress curl (relation 4.1 in Zhang and Valls, 2007) is constant, which implies that the local vorticity balance is primarily influenced by changes in bottom vortex stretching.

Shall we understand this equation from a barotropic perspective, or shall we (also) consider the question from a baroclinic perspective, as the freshwater

input can alter ocean stratification and hence affect the topography steering (e.g., Marshall 1995)? In another word, this could be somehow helpful to understand the relationship between the Gulf Stream and the AMOC, since the Gulf Stream was considered to include both wind-driven (but fixed wind in the experiment?) and thermohaline components.

There are indeed baroclinic responses across the GS front, visualised by the different isopycnal slopes in Figures 4a,b and S4. This is primarily induced by the thermohaline component, and not so much by the applied freshwater flux forcing; the hosing is distributed over a large region (20°N – 50°N). Below 1,000 m depths, there are hardly any changes in the stratification and the baroclinic responses are relatively small there. This baroclinic contribution, including the topographic steering effects, are now discussed in the revision.

Additionally, I still would like to highlight the importance of topography, as the bottom pressure torque (BPT) appears as a dominant term in the vorticity balance of the AMOC, representing a key dynamical link between the overturning and gyre streamfunctions in the North Atlantic (Yeager 2015).

This is indeed a relevant reference, which is now included in the revised manuscript.

As related, the authors may want to notice the effect of surface winds as their winds seem to be fixed in the freshwater hosing experiment in ocean-alone model. In a fully coupled climate model, a weakened AMOC can generate anticyclonic surface wind anomalies over the subpolar North Atlantic to decelerate the subpolar gyre circulation, together with a weaker northward Gulf Stream (Mimi and Liu 2024). Atmospheric wind response to AMOC change, which further influences the Gulf Stream, needs to be discussed from the perspective of ocean-atmosphere coupling.

The surface wind forcing is indeed fixed in the POP, hence this cannot be tested in our simulation. This is made more explicit in the revised Introduction. The findings by Mimi and Liu (2024) are indeed relevant for the coupled case, which are included in the revised Discussion.

Moreover, the authors mentioned in Lines 124-126, Page 4 that “What is

most relevant is that this GS path transition occurs approximately 25 years prior to the onset of the AMOC tipping event. The different GS paths may be related to different GS path equilibria“. Can the authors elaborate on the cause of the timescale, in particular, why it should be 25 years?

It is not entirely clear where this 25-year lead-lag timescale originates from, hence we omitted this sentence in the revision. The timescale could be related to a certain bottom vorticity threshold (lines 188 – 190), but this analysis is out of scope. We discuss the 25-year timescale in greater detail in the revised Discussion.

Besides, since the high-resolution POP is used, I am wondering whether the well-simulated oceanic mesoscale eddies will allow for a better representation of eddy momentum fluxes (ref. 7, Greatbatch et al. 2010) and eddy–mean flow interactions (Kang and Curchitser 2015) in the Gulf Stream Region.

Explicitly representing mesoscale eddies improves eddy–mean flow interactions in the Gulf Stream region, but this is beyond the scope of this study. We determined the GS destabilisation point in the revised manuscript (suggested by Reviewer #1), which agrees well with that of observations. This GS destabilisation point cannot be determined for the low-resolution POP, indirectly implying that eddy–mean flow interactions improve the (down-stream) GS dynamics. We have included the two references above in the revised Introduction.

MS-No.: COMMSENV-25-4192A

Version: Revision II

Title: Abrupt Gulf Stream Path Changes are a Precursor to an AMOC Collapse

Author(s): René M. van Westen and Henk A. Dijkstra

Point-by-point reply to reviewers

January 15, 2026

We thank the reviewers again for their careful reading and for the useful comments on the revised manuscript.

Reviewer #1

2nd review of van Westen and Dijkstra.

I really like this paper, it presents a convincing study of Gulf Stream (GS) shifts in response to AMOC collapse. The inclusion of observational data is very plausible. I think the extended observational GS north wall strengthens the analysis. I was going to suggest to use the Taylor GS index rather than Joyce to avoid the warming problem but it looks like that isn't available any longer. I am happy that the authors have addressed my comments and can recommend the paper for publication..

Minor comments:

1. *l37, suggest replacing 'interesting' with a more quantitative word*
We replaced 'interesting' by 'noteworthy'.
2. *l54, suggest diluting the claim or specifying that all the studies cited (14, 34, 35) use SST as an AMOC proxy. Adding "...reconstructions *based on SST data* suggest..."*
Suggestion followed.
3. *There are outstanding formatting issues in the references such as incorrect capitalisation (LaTeX issue!) Taylor, A., & Stephens, J. (1980). Latitudinal displacements of the*

Gulf-Stream (1966 to 1977) and their relation to changes in temperature and zooplankton abundance in the NE Atlantic. Oceanologica Acta, 3(2), 145-149.

The reference list will be changed to the in-house journal style.

Reviewer #2

I believe all relevant remarks have been addressed and this MS can be published. Thank you

Thank you for the re-review.

Reviewer #3

I am generally fine with the authors' reply and revision, and thus recommend acceptance.

Thank you for the re-review.